# HIDDEN MARKOV MODELS ARE RECURRENT NEURAL NETWORKS: A DISEASE PROGRESSION MODELING APPLICATION

## ABSTRACT

Hidden Markov models (HMMs) are commonly used for disease progression modeling when the true patient health state is not fully known. Since HMMs may have multiple local optima, performance can be improved by incorporating additional patient covariates to inform estimation. To allow for this, we formulate a special case of recurrent neural networks (RNNs), which we name hidden Markov recurrent neural networks (HMRNNs), and prove that each HMRNN has the same likelihood function as a corresponding discrete-observation HMM. The HMRNN can be combined with any other predictive neural networks that take patient covariate information as input. We show that HMRNN parameter estimates are numerically close to those obtained from via the Baum-Welch algorithm, validating their theoretical equivalence. We then demonstrate how the HMRNN can be combined with other neural networks to improve parameter estimation, using an Alzheimer's disease dataset. The HMRNN's solution improves disease forecasting performance and offers a novel clinical interpretation compared with a standard HMM.

## 1 INTRODUCTION

Hidden Markov models (HMMs; Baum & Petrie, 1966) are commonly used for modeling disease progression, because they allow researchers to conceptualize complex (and noisy) clinical measurements as originating from a smaller set of latent health states. Each latent health state is characterized by an emission distribution that specifies the probabilities of each measurement/observation given that state. This allows HMMs to explicitly account for uncertainty or measurement error, since the system's true state is not fully observable. Because of their intuitive parameter interpretations and flexibility, HMMs have been used to model biomarker changes in HIV patients (Guihenneuc-Jouyaux et al., 2000), Alzheimer's disease progression (Liu et al., 2015), breast cancer screening decisions (Ayer et al., 2012), and patient response to blood anticoagulants (Nemati et al., 2016).

Researchers may wish to integrate HMMs with other disease progression models and/or data sources. For instance, researchers in Igl et al. (2018) jointly trained parameters for an HMM and a reinforcement learning policy to maximize patient returns. Other researchers have attempted to learn or initialize HMM parameters based on additional sources of patient data (Gupta, 2019; Zhou et al., 2019). Such modifications typically require multiple estimation steps (e.g., Zhou et al., 2019) or changes to parameter interpretation (e.g., Igl et al., 2018). This is because the standard algorithm for fitting HMMs, the Baum-Welch algorithm (Baum & Petrie, 1966), maximizes the likelihood of a data sequence without consideration of additional covariates.

We introduce Hidden Markov Recurrent Neural Networks (HMRNNs) - neural networks that mimic the computation of hidden Markov models while allowing for substantial modularity with other predictive networks. Unlike past work combining neural networks and HMMs (e.g., Bridle, 1990), HMRNNs are designed to maximize the most commonly-used HMM fit criterion - the likelihood of the data given the parameters. In doing so, our primary contributions are as follows: (1) We prove how recurrent neural networks (RNNs) can be formulated to optimize the same likelihood function as HMMs, with parameters that can be interpreted as HMM parameters (section 3); (2) We empirically demonstrate that our model yields statistically similar parameter solutions compared with the Baum-

Welch algorithm (section 4.1); (3) We demonstrate our model's utility in a disease progression application, in which combining it with other predictive neural networks improves predictive accuracy and offers unique parameter interpretations not afforded by simple HMMs (section 4.2).

## 2  RELATED WORK

A small number of studies have attempted to formally model HMMs in a neural network context. Wessels & Omlin (2000) proposes using neural networks to approximate Gaussian emission distributions in HMMs; however, their method requires pre-training of the HMM. Similar to our work, Bridle (1990) demonstrates how HMMs can be reduced to recurrent neural networks for speech recognition, though it requires that neurons be computed via products (rather than sums), which are not commonly used in modern neural networks. This model also maximizes the mutual information between observations and hidden states; this is a commonly used criterion in speech recognition, but less common than likelihood maximization in other domains (e.g., disease progression modeling). Lastly, Bridle (1990) and Wessels & Omlin (2000) present only theoretical justification, with no empirical comparisons with the Baum-Welch algorithm.

A limited number of studies have also explored connections between neural networks and Markov models in the healthcare domain. For instance, Nemati et al. (2016) employs a discriminative hidden Markov model to estimate 'hidden states' underlying patients' ICU measurements, though these hidden states are not mathematically equivalent to HMM latent states. Estebanez et al. (2012) compares HMM and neural network effectiveness in training a robotic surgery assistant, while Baucum et al. (2020) proposes a generative neural network for modeling ICU patient health based on the mathematical intuition of the HMM. Although these studies showcase the of value of pairing neural networks and Markov models in the healthcare domain, they differ from our approach of directly formulating HMMs as neural networks, which maintains the interpretability of HMMs while allowing for joint estimation of the HMM with other predictive models.

In summary, studies have shown the promise of incorporating elements of HMMs into deep learning tasks, there are no existing methods for optimizing HMM log-likelihood in a neural network context. While past works have also used gradient descent to learn HMM parameters (e.g. Yildirim et al., 2015), we demonstrate how specifically implementing HMMs as *neural networks* allows additional data sources (e.g., patient covariates) to steer model estimation to better-fitting solutions. We thus develop the first neural network formulation of an HMM that maximizes the observed data likelihood, employs widely-used neural network operations, and compares favorably to the Baum-Welch algorithm when tested on real-world datasets.

## 3  METHODS

In this section, we briefly review HMM preliminaries, formally define the HMRNN, and prove that it optimizes the same likelihood function as a corresponding HMM.

### 3.1  HMM PRELIMINARIES

Formally, an HMM models a system over a given time horizon $T$, where the system occupies a hidden state $x_t \in S = \{1, 2, \ldots, k\}$ at any given time point $t \in \{0, 1, \ldots, T\}$; that is, $x_t = i$ indicates that the system is in the $i$th state at time $t$. For any state $x_t \in S$ and any time point $t \in \{0, 1, \ldots, T\}$, the system emits an observation according to an emission distribution that is uniquely defined for each state. We consider the case of categorical emission distributions, which are commonly used in healthcare (e.g., Liu et al., 2015; Leon, 2015; Ayer et al., 2012; Stanculescu et al., 2013). These systems emit one of $c$ distinct observations at each time point; that is, for any time $t$, we observe $y_t \in O$, where $|O| = c$ and $O = \{1, \ldots, c\}$.

Thus, an HMM is uniquely defined by a $k$-length initial probability vector $\boldsymbol{\pi}$, $k \times k$ transition matrix $\boldsymbol{P}$, and $k \times c$ emission matrix $\boldsymbol{\Psi}$. Entry $i$ in the vector $\boldsymbol{\pi}$ is the probability of starting in state $i$, row $i$ in the matrix $\boldsymbol{P}$ is the state transition probability distribution from state $i$, and row $i$ of the matrix $\boldsymbol{\Psi}$ is the emission distribution from state $i$.

HMMs are fit via the Baum-Welch algorithm, which identifies the parameters that (locally) maximize the likelihood of the observed data (Baum & Petrie, 1966). The likelihood of an observation sequence $\boldsymbol{y}$ is a function of an HMM's initial state distribution ($\boldsymbol{\pi}$), transition probability matrix ($\boldsymbol{P}$), and emission matrix ($\boldsymbol{\Psi}$) (Jurafsky & Martin, 2009). Let $\mathrm{diag}(\boldsymbol{\Psi}_i)$ be a $k \times k$ diagonal matrix with the $i$th column of $\boldsymbol{\Psi}$ as its entries - i.e., the probabilities of observation $i$ from each state. We then have

$$\mathrm{Pr}(\boldsymbol{y}|\boldsymbol{\pi}, \boldsymbol{P}, \boldsymbol{\Psi}) = \boldsymbol{\pi}^\top \cdot \mathrm{diag}(\boldsymbol{\Psi}_{y_0}) \cdot (\prod_{i=1}^{T} \boldsymbol{P} \cdot \mathrm{diag}(\boldsymbol{\Psi}_{y_i})) \cdot \mathbf{1}_{k \times 1}. \tag{1}$$

The likelihood function can also be expressed in terms of $\alpha_t(i)$, the probability of being in state $i$ at time $t$ *and* having observed $\{y_0, ..., y_t\}$. We denote $\boldsymbol{\alpha}_t$ as the (row) vector of all $\alpha_t(i)$ for $i \in S$, with

$$\boldsymbol{\alpha}_t = \boldsymbol{\pi}^\top \cdot \mathrm{diag}(\boldsymbol{\Psi}_{y_0}) \cdot (\prod_{i=1}^{t} \boldsymbol{P} \cdot \mathrm{diag}(\boldsymbol{\Psi}_{y_i})) \tag{2}$$

for $t \in \{1, ..., T\}$, with $\boldsymbol{\alpha}_0 = \boldsymbol{\pi}^\top \cdot \mathrm{diag}(\boldsymbol{\Psi}_{y_0})$. Note that equation 3.1 also implies that $\boldsymbol{\alpha}_t = \boldsymbol{\alpha}_{t-1} \cdot \boldsymbol{P} \cdot \mathrm{diag}(\boldsymbol{\Psi}_{y_t})$ for $t \in \{1, ..., T\}$, and that

$$\mathrm{Pr}(\boldsymbol{y}|\boldsymbol{\pi}, \boldsymbol{P}, \boldsymbol{\Psi}) = \boldsymbol{\alpha}_T \cdot \mathbf{1}_{k \times 1}. \tag{3}$$

## 3.2 Definition of hidden Markov recurrent neural networks (HMRNNs)

An HMRNN is a recurrent neural network whose parameters directly correspond to the initial state, transition, and emission probabilities of an HMM. As such, training an HMRNN optimizes the joint log-likelihood of the $N$ $T$-length observation sequences given these parameters.

**Definition 3.1.** *An HMRNN is a recurrent neural network with trainable parameters $\boldsymbol{\pi}$ (a $k$-length stochastic vector), $\boldsymbol{P}$ (a $k \times k$ stochastic matrix), and $\boldsymbol{\Psi}$ (a $k \times c$ stochastic matrix). It is trained on $T + 1$ input matrices of size $N \times c$, denoted by $\boldsymbol{Y}_t$ for $t \in \{0, 1, \ldots, T\}$, where the $n$-th row of matrix $\boldsymbol{Y}_t$ is a one-hot encoded vector of observation $y_t^{(n)}$ for sequence $n \in \{1, \ldots, N\}$. The HMRNN consists of an inner block of hidden layers that is looped $T + 1$ times (for $t \in \{0, 1, \ldots, T\}$), with each loop containing hidden layers $\boldsymbol{h}_1^{(t)}$, $\boldsymbol{h}_2^{(t)}$, and $\boldsymbol{h}_3^{(t)}$, and a $c$-length input layer $\boldsymbol{h}_y^{(t)}$ through which the input matrix $\boldsymbol{Y}_t$ enters the model. The HMRNN has a single output unit $o^{(T)}$ whose value is the joint negative log-likelihood of the $N$ observation sequences under an HMM with parameters $\boldsymbol{\pi}$, $\boldsymbol{P}$, and $\boldsymbol{\Psi}$; the summed value of $o^{(T)}$ across all $N$ observation sequences is also the loss which is minimized through any neural network optimizer (e.g., gradient descent).*

*Layers $\boldsymbol{h}_1^{(t)}$, $\boldsymbol{h}_2^{(t)}$, $\boldsymbol{h}_3^{(t)}$, and $o^{(T)}$ are defined in the following equations. Note that the block matrix in equation (5) is a $c \times (kc)$ block matrix of $c$ $\mathbf{1}_{1 \times k}$ vectors , arranged diagonally, while the block matrix in equation (6) is a $(kc) \times k$ row-wise concatenation of $c$ $k \times k$ identity matrices.*

$$\boldsymbol{h}_1^{(t)} = \begin{cases} \boldsymbol{\pi}^\top, & t = 0, \\ \boldsymbol{h}_3^{(t-1)} \boldsymbol{P}, & t > 0. \end{cases} \tag{4}$$

$$\boldsymbol{h}_2^{(t)} = \mathrm{ReLu}\left(\boldsymbol{h}_1^{(t)} [\mathrm{diag}(\boldsymbol{\Psi}_1) \ldots \mathrm{diag}(\boldsymbol{\Psi}_c)] + \boldsymbol{Y}_t \begin{bmatrix} \mathbf{1}_{1 \times k} & \ldots & \mathbf{0}_{1 \times k} \\ \ldots & \ldots & \ldots \\ \mathbf{0}_{1 \times k} & \ldots & \mathbf{1}_{1 \times k} \end{bmatrix} - \mathbf{1}_{n \times (kc)}\right) \tag{5}$$

$$\boldsymbol{h}_3^{(t)} = \boldsymbol{h}_2^{(t)} [\boldsymbol{I}_k \quad \ldots \quad \boldsymbol{I}_k]^\top \tag{6}$$

$$o^{(T)} = -\log(\boldsymbol{h}_3^{(T)} \mathbf{1}_{k \times 1}). \tag{7}$$

Fig. 1 outlines the structure of the HMRNN. Intuitively, operations within each recurrent block mimic matrix multiplication by $\mathrm{diag}(\boldsymbol{\Psi}_{y_t})$ (i.e., $\boldsymbol{h}_3^{(t)} = \boldsymbol{h}_1^{(t)} \mathrm{diag}(\boldsymbol{\Psi}_{y_t})$), while connections between blocks mimic multiplication by $\boldsymbol{P}$. In each block, layer $\boldsymbol{h}_1^{(t)}$ contains $k$ units that represent the joint probability of being in each state $1$–$k$ and all observations up to time $t$ - note that this is equivalent to the $\alpha$ values in traditional HMM notation. Layer $\boldsymbol{h}_2^{(t)}$, expands each unit in $\boldsymbol{h}_1^{(t)}$ into $c$ units via connections with weights $\boldsymbol{\Psi}_{i,j}$ for $i \in \{1, \ldots, k\}$ and $j \in \{1, \ldots, c\}$, resulting in $k \cdot c$ units; this is equivalent to multiplying $\boldsymbol{h}_1^{(t)}$ (in row-vector form) by a column-wise concatenation of $\mathrm{diag}(\boldsymbol{\Psi}_j)$

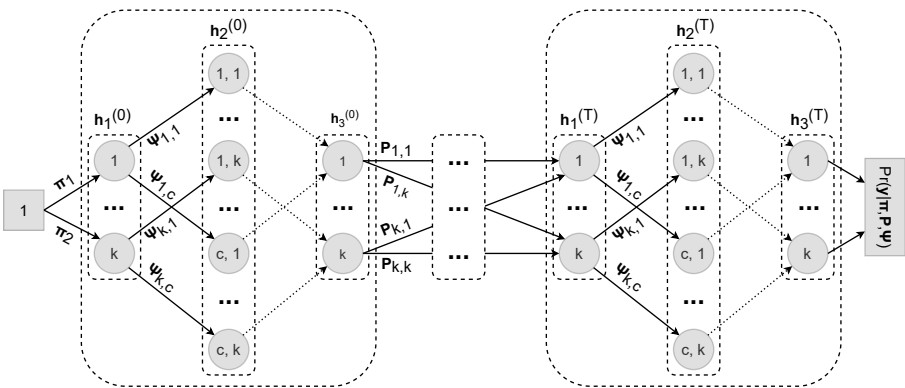

Figure 1: Structure of the hidden Markov recurrent neural network (HMRNN). Solid lines indicate learned weights that correspond to HMM parameters; dotted lines indicate weights fixed to 1. The inner block initializes with the initial state probabilities then mimics multiplication by $\operatorname{diag}(\boldsymbol{\Psi}_{y_t})$; connections between blocks mimic multiplication by $\boldsymbol{P}$.

for $j \in \{1, \ldots, c\}$. The resulting units represent the probabilities of all possible state/outcome combinations at time $t$.

Each column in $\boldsymbol{Y}_t$ is connected to all units in $\boldsymbol{h}_2^{(t)}$ that correspond to that column's observation, with connection weights set to 1. A bias of $-1$ and subsequent ReLu activation are then applied to layer $\boldsymbol{h}_2^{(t)}$; this leaves the units that correspond to $\boldsymbol{Y}_t$ unchanged, while all other units (i.e., probabilities for non-occurring observations) are made negative by the $-1$ bias, then forced to zero through the ReLu activation. Thus, layer $\boldsymbol{h}_2^{(t)}$ identifies the joint probability of being in each state *and* observing $\boldsymbol{Y}_t$. Layer $\boldsymbol{h}_3^{(t)}$ then sums across all $c$ units for each of the $k$ states (all of which are zero except for those corresponding to $\boldsymbol{Y}_t$), yielding $k$ unitsrepresenting the probabilities of being in each state given all previous observations *and* the observation at time $t$, i.e., $\boldsymbol{h}_3^{(t)} = \boldsymbol{h}_1^{(t)} \operatorname{diag}(\boldsymbol{\Psi}_{y_t})$.

We then apply a fully-connected layer of weights to transform $\boldsymbol{h}_3^{(3)}$ to $\boldsymbol{h}_1^{(t+1)}$, equivalent to matrix multiplication by $\boldsymbol{P}$, i.e., $\boldsymbol{h}_1^{(t+1)} = \boldsymbol{h}_3^{(t)} \boldsymbol{P}$, forcing the rows of $\boldsymbol{P}$ to sum to 1. Lastly, activations for $\boldsymbol{h}_3^{(T)}$ are summed into unit $o^{(T)}$ and subject to a negative logarithmic activation function, yielding the negative log-likelihood of the data given the model's parameters. Note that the hidden layers may suffer from underflow for long sequences. This can be addressed by normalizing layer $\boldsymbol{h}_3^{(t)}$ to sum to 1 at each time point, then simply subtracting the logarithm of the normalization term (i.e., the log-sum of the activations) from network's output $-\log(o^{(T)})$.

The HMRNN is a special case of an RNN, due to its time-dependent layers and shared weights between time points. Note that its use of recurrent blocks (each of which containing three layers) differs from many RNNs that use only a single layer at each time point; this distinction allows the HMRNN to mimic HMM computations of an HMM at each time point.

### 3.3 PROOF OF HMM/HMRNN EQUIVALENCE

We now formally establish that the HMRNN's output unit, $o^{(T)}$, is the negative log-likelihood of an observation sequence under an HMM with parameters $\boldsymbol{\pi}$, $\boldsymbol{P}$, and $\boldsymbol{\Psi}$. We prove this for the case of $N = 1$ and drop notational dependence on $n$ (i.e., we write $y_t^{(1)}$ as $y_t$), though extension to $N > 1$ is trivial since the log-likelihood of multiple independence sequences is simply the sum of their individual log-likelihoods. We first rely on the following lemma.

**Lemma 3.1.** *If all units in $\boldsymbol{h}_1^{(t)}(j)$ are between 0 and 1 (inclusive), then $\boldsymbol{h}_3^{(t)} = \boldsymbol{h}_1^{(t)} \operatorname{diag}(\boldsymbol{\Psi}_{y_t})$.*

*Proof.* Let $\boldsymbol{h}_1^{(t)}(j)$ and $\boldsymbol{h}_3^{(t)}(j)$ represent the $j$th units of layer $\boldsymbol{h}_1^{(t)}$ and $\boldsymbol{h}_3^{(t)}$, respectively. Showing $\boldsymbol{h}_3^{(t)} = \boldsymbol{h}_1^{(t)} \operatorname{diag}(\boldsymbol{\Psi}_{y_t})$ is equivalent to showing that $\boldsymbol{h}_3^{(t)}(j) = \boldsymbol{\Psi}_{j,y_t} \boldsymbol{h}_1^{(t)}(j)$ for $j \in \{1, .., k\}$.

To show this, recall that $\boldsymbol{h}_2^{(t)}$ contains $k \times c$ units, which we index with a tuple $(l, m)$ for $l \in \{1, \ldots, c\}$ and $m \in \{1, \ldots, k\}$. The connection matrix between $\boldsymbol{h}_1^{(t)}$ and $\boldsymbol{h}_2^{(t)}$ is $[\, \mathrm{diag}(\boldsymbol{\Psi}_1) \ldots \mathrm{diag}(\boldsymbol{\Psi}_c)]$. Thus, the connection between units $\boldsymbol{h}_1^{(t)}(j)$ and $\boldsymbol{h}_2^{(t)}(l, m)$ is $\boldsymbol{\Psi}_{j,l}$ when $j = m$, and equals 0 otherwise. Also recall that matrix $\boldsymbol{Y}_t$ enters the model through a $c$-length input layer $\boldsymbol{h}_y^{(t)}$, where the $j$th unit is 1 when $y_t = j$, and equals 0 otherwise. This layer is connected to $\boldsymbol{h}_2^{(t)}$ by a $c \times (kc)$ diagonal block matrix of $c$ $(1 \times k)$ row vectors of ones. Thus, the connection between the $j$th unit of this input layer and unit $(l, m)$ of $\boldsymbol{h}_2^{(t)}$ is 1 when $j = l$, and equals 0 otherwise. Lastly, a bias of $-1$ is added to all units in $\boldsymbol{h}_2^{(t)}$, which is then subject to a ReLu activation, resulting in the following expression for each unit in $\boldsymbol{h}_2^{(t)}$:

$$\boldsymbol{h}_2^{(t)}(l, m) = \mathrm{ReLu}(\boldsymbol{\Psi}_{m,l} \cdot \boldsymbol{h}_1^{(t)}(m) + \boldsymbol{h}_y^{(t)}(l) - 1). \tag{8}$$

Because $\boldsymbol{h}_y^{(t)}(l)$ is 1 when $y_t = l$, and equals 0 otherwise, then if all units in $\boldsymbol{h}_1^{(t)}$ are between 0 and 1, we have

$$\boldsymbol{h}_2^{(t)}(l, m) = \begin{cases} \mathrm{ReLu}(\boldsymbol{\Psi}_{m,l} \cdot \boldsymbol{h}_1^{(t)}(m)) = \boldsymbol{\Psi}_{m,l} \cdot \boldsymbol{h}_1^{(t)}(m), & j = y_t, \\ \mathrm{ReLu}(\boldsymbol{\Psi}_{m,l} \cdot \boldsymbol{h}_1^{(t)}(m) - 1) = 0, & \text{otherwise} . \end{cases} \tag{9}$$

The connection matrix between $\boldsymbol{h}_2^{(t)}$ and $\boldsymbol{h}_3^{(t)}$ is a $(kc) \times k$ row-wise concatenation of $k \times k$ identity matrices; thus, the connection between $\boldsymbol{h}_2^{(t)}(l, m)$ and $\boldsymbol{h}_3^{(t)}(j)$ is 1 if $j = m$, and 0 otherwise. Hence,

$$\boldsymbol{h}_3^{(t)}(j) = \sum_{j=0}^{c} \boldsymbol{h}_2^{(t)}(l, j) = \boldsymbol{\Psi}_{j,y_t} \cdot \boldsymbol{h}_1^{(t)}(j). \tag{10}$$

$\square$

Thus, $\boldsymbol{h}_3^{(t)} = \boldsymbol{h}_1^{(t)} \mathrm{diag}(\boldsymbol{\Psi}_{y_t})$.

**Theorem 3.1.** *An HMRNN with parameters $\boldsymbol{\pi}$ ($1 \times k$ stochastic vector), $\boldsymbol{P}$ ($k \times k$ stochastic matrix), and $\boldsymbol{\Psi}$ ($k \times c$ stochastic matrix), and with layers defined as in equations (4-7), produces output neuron $o^{(T)}$ for sequence $n \in \{1, \ldots, N\}$ whose value is the negative log-likelihood of a corresponding HMM as defined in equation (1).*

*Proof.* Note that, based on Lemma 3.1 and equation 4, $\boldsymbol{h}_3^{(t)} = \boldsymbol{h}_3^{(t-1)} \cdot \boldsymbol{P} \cdot \mathrm{diag}(\boldsymbol{\Psi}_{y_t})$ for $t \in \{1, ..., T\}$, assuming that $\boldsymbol{h}_1^{(t)}(j) \in [0, 1]$ for $j \in \{1, .., k\}$. Since $\boldsymbol{\alpha}_t = \boldsymbol{\alpha}_{t-1} \cdot \boldsymbol{P} \cdot \mathrm{diag}(\boldsymbol{\Psi}_{y_t})$, then if $\boldsymbol{h}_3^{(t-1)} = \boldsymbol{\alpha}_{t-1}$, then $\boldsymbol{h}_1^{(t)}(j) \in [0, 1]$ for $j \in \{1, .., k\}$ and therefore $\boldsymbol{h}_3^{(t)} = \boldsymbol{\alpha}_t$. We show the initial condition that $\boldsymbol{h}_3^{(0)} = \boldsymbol{\alpha}_0$, since $\boldsymbol{h}_1^{(0)} = \boldsymbol{\pi}^\top$ implies that $\boldsymbol{h}_3^{(0)} = \boldsymbol{\pi}^\top \cdot \mathrm{diag}(\boldsymbol{\Psi}_{y_0}) = \boldsymbol{\alpha}_0$. Therefore, by induction, $\boldsymbol{h}_3^{(T)} = \boldsymbol{\alpha}_T$, and $o^{(T)} = -\log(\boldsymbol{\alpha}_T \cdot \boldsymbol{1}_{k \times 1})$, which is the logarithm of the HMM likelihood from equation 3. $\square$

## 4 EXPERIMENTS AND RESULTS

In this section, we compare HMRNNs to Baum-Welch through computational experiments with synthetic data and a case study of Alzheimer's disease patients. The experiment in section 4.1 demonstrates that Baum-Welch and and the HMRNN yield similar parameter solutions and predictive accuracy. Section 4.2 demonstrates how augmenting the HMRNN with additional neural networks yields HMM parameters with *improved* predictive ability on real-world data.

### 4.1 EMPIRICAL VALIDATION OF HMRNN

We demonstrate that an HMRNN trained via gradient descent yields statistically similar solutions to the Baum-Welch algorithm. We show this with synthetically-generated observations sequences for which the true HMM parameters are known, allowing us to validate Theorem 3.1 empirically.

We simulate systems with state spaces $S = 1, 2, ..., k$ that begin in state 1, using $k = 5$, 10, or 20 states. These state sizes are consistent with disease progression HMMs, which often involve less than 10 states (Zhou et al., 2019; Jackson et al., 2003; Sukkar et al., 2012; Martino et al., 2020). We assume that each state 'corresponds' to one observation, implying the same number of states and observations ($c = k$). The probability of correctly observing a state ($P(y_t = x_t)$) is $\psi_{ii}$, which is the diagonal of $\mathbf{\Psi}$ and is the same for all states. We simulate systems with $\psi_{ii} = 0.6, 0.75$, and 0.9.

We test three variants of the transition probability matrix $\mathbf{P}$. Each is defined by their same-state transition probability $p_{ii}$, which is the same for all states. For all $\mathbf{P}$ the probability of transitioning to higher states increases with state membership; this is known as 'increasing failure rate' and is a common property for Markov processes. As $p_{ii}$ decreases, the rows of $\mathbf{P}$ stochastically increase, i.e., lower values of $p_{ii}$ imply a greater chance of moving to higher states. We use values of $p_{ii} = 0.4, 0.6$, and 0.8, for 27 total simulations ($k = \{5, 10, 20\} \times \mathbf{\Psi}_{ii} = \{0.6, 0.75, 0.9\} \times p_{ii} = \{0.4, 0.6, 0.8\}$).

For each of the 27 simulations, we generate 100 trajectories of length $T = 60$; this time horizon might practically represent one hour of data collected each minute or two months of data collected each day. Initial state probabilities are fixed at 1 for state 1 and 0 otherwise. Transition parameters are initialized based on the observed number of transitions in each dataset, using each observation as a proxy for its corresponding state. Since transition probabilities are initialized assuming no observation error, the emission matrices are correspondingly initialized using $\psi_{ii} = 0.95$ (with the remaining 0.05 distributed evenly across all other states). In practice, gradient descent on the HMRNN rarely yields parameter values that are exactly zero. To facilitate comparability between the HMRNN and Baum-Welch, we post-process all HMRNN results with one iteration of the Baum-Welch algorithm, which forces low-probability entries to zero. For Baum-Welch and HMRNN, training ceased when all parameters ceased to change by more than 0.001. For each simulation, we compare Baum Welch's and the HMRNN's average Wasserstein distance between the rows of the estimated and ground truth $\mathbf{P}$ and $\mathbf{\Psi}$ matrices. This serves as a measure of each method's ability to recover the true data-generating parameters. We also compare the Baum-Welch and HMRNN solutions' log-likelihoods using a separate hold-out set of 100 trajectories.

Across all simulations, the average Wasserstein distance between the rows of the true and estimated transition matrices was 0.191 for Baum-Welch and 0.178 for HMRNN (paired $t$-test $p$-value of 0.483). For the emission matrices, these distances were 0.160 for Baum-Welch and 0.137 for HMRNN (paired $t$-test $p$-value of 0.262). This suggests that Baum-Welch and the HMRNN recovered the ground truth parameters with statistically similar degrees of accuracy. This can be seen in Figure 2, which presents the average estimated values of $p_{ii}$ and $\psi_{ii}$ under each model. Both models' estimated $p_{ii}$ values are, on average, within 0.05 of the ground truth values, while they tended to estimate $\psi_{ii}$ values of around 0.8 regardless of the true $\psi_{ii}$. Note that, while Baum-Welch was slightly more accurate at estimating $p_{ii}$ and $\psi_{ii}$, the overall distance between the ground truth and estimated parameters did not significantly differ between Baum-Welch and the HMRNN.

For each simulation, we also compute the log-likelihood of a held-out set of 100 sequences under the Baum-Welch and HMRNN parameters, as a measure of model fit. The average holdout log-likelihoods under the ground truth, Baum-Welch, and HMRNN parameters are -9250.53, -9296.03, and -9303.27, respectively (paired $t$-test $p$-value for Baum-Welch/HMRNN difference of 0.440). Thus, Baum-Welch and HMRNN yielded similar degrees of model fit on held-out sequence data.

## 4.2 Alzheimer's disease symptom progression application

We also demonstrate how incorporating additional data into an HMRNN can improve parameter fit and offer novel clinical interpretations. We test our HMRNN on clinical data from $n = 426$ patients with mild cognitive impairment (MCI), collected over the course of three ($n = 91$), four ($n = 106$), or five ($n = 229$) consecutive annual clinical visits (Initiative). Given MCI patients' heightened risk of Alzheimer's, modeling their symptom progression is of considerable clinical interest (Hirao et al., 2005; Hansson et al., 2010; Rabin et al., 2009). We analyze patients' overall cognitive functioning based on the Mini Mental Status Exam (MMSE; Folstein et al., 1975).

MMSE scores range from 0 to 30, with scores of $27 - 30$ indicating no cognitive impairment, $24 - 26$ indicating borderline cognitive impairment, and $17 - 23$ indicating mild cognitive impairment (Chopra et al., 2007; Monroe & Carter, 2012). Scores below 17 were infrequent (1.2%) and were treated as scores of 17 for analysis. We use a 3-state latent space $S = \{0, 1, 2\}$, with $x_t = 0$ representing

Estimated vs. Ground Truth Parameters, Baum-Welch vs. HMRNN

Figure 2: Estimated $p_{ii}$ (left) and $\psi_{ii}$ (right) under Baum-Welch and HMRNN, shown by ground truth parameter value. Results for each column are averaged across 9 simulations. Dashed lines indicate ground truth $p_{ii}$ (left) and $\psi_{ii}$ (right) values, and error bars indicate 95% confidence intervals (but do not represent tests for significant differences). In line with Theorem 3.1, Baum-Welch and the HMRNN produce near-identical parameter solutions according to the Wasserstein distance metric.

'no cognitive impairment,' $x_t = 1$ representing 'borderline cognitive impairment,' and $x_t = 2$ representing 'mild cognitive impairment.' The observation space is $O = \{0, 1, 2\}$, using $y_t = 0$ for scores of $27 - 30$, $y_t = 1$ for scores of $24 - 26$, and $y_t = 2$ for scores of $17 - 23$.

To showcase the benefits of the HMRNN's flexibility, we augment the HMRNN through two substantive modifications. First, the initial state probabilities in the augmented HMRNN are predicted from patients' gender, age, degree of temporal lobe atrophy (Hua et al., 2008), and amyloid-beta 42 levels (A$\beta$42, a relevant Alzheimer's biomarker (Canuet et al., 2015; Blennow, 2004)), using a single-layer neural network. Second, at each time point, the probability of being in the most impaired state, $\boldsymbol{h}_t^{(1)}(2)$, is used to predict concurrent scores on the Clinical Dementia Rating (CDR, Morris, 1993), a global assessment of dementia severity, allowing another clinical metric to inform estimation. We use a single connection and sigmoid activation to predict patients' probability of receiving a CDR score above 0.5 (corresponding to 'mild dementia').

We compare HMRNN and Baum-Welch parameter solutions' ability to predict patients' final MMSE score categories from their initial score categories, using 10-fold cross-validation. We evaluate performance using weighted log-loss $L$ (Guerrero-Pena et al., 2018; Stelmach & Chlebus, 2020), i.e., the log-probability placed on each final MMSE score category averaged across score categories. This metric accounts for class imbalance and rewards models' confidence in their predictions, an important component of medical decision support (Bussone et al., 2015; Lim & Dey, 2010). We also report $\bar{p}$, the average probability placed on patients' final MMSE scores (computed from $L$). We train all models using a relative log-likelihood tolerance of $0.001\%$ (we do not use parameter convergence since the number of parameters differs between models). Converge runtimes for Baum-Welch and the HMRNN are 2.89 seconds and 15.24 seconds, respectively.

Model results appear in Table 1. Note that the HMRNN's weighted log-loss $L$ is significantly lower than Baum-Welch's (paired $t$-test $p$-value$= 2.396 \times 10^{-6}$), implying greater predictive performance. This is supported by Figure 3, which shows $\bar{p}$, the average probability placed on patients' final MMSE scores by score category. Note that error bars represent marginal sampling error and do not represent statistical comparisons between Baum-Welch and HMRNN. Interestingly, the HMRNN yields lower transition probabilities and lower MMSE accuracy (i.e., lower diagonal values of $\boldsymbol{\Psi}$) than Baum-Welch, suggesting that score changes are more likely attributable to testing error as opposed to true state changes.

## 5 DISCUSSION

We outline a flexible approach for HMM estimation using neural networks. The HMRNN produces statistically similar solutions to the Baum-Welch algorithm in its standard form, yet can be combined

Table 1: Results from Alzheimer's disease case study.

|  | Baum-Welch | | | HMRNN | | |
|---|---|---|---|---|---|---|
| $\pi$ | 0.727 | 0.271 | 0.002 | 0.667 | 0.333 | 0.000 |
| $P$ | 0.898 | 0.080 | 0.022 | 0.970 | 0.028 | 0.002 |
|  | 0.059 | 0.630 | 0.311 | 0.006 | 0.667 | 0.327 |
|  | 0.000 | 0.016 | 0.984 | 0.000 | 0.003 | 0.997 |
| $\Psi$ | 0.939 | 0.060 | 0.001 | 0.930 | 0.067 | 0.003 |
|  | 0.175 | 0.819 | 0.006 | 0.449 | 0.548 | 0.003 |
|  | 0.004 | 0.160 | 0.836 | 0.005 | 0.308 | 0.687 |
| $L$ | | 0.991 | | | 0.884 | |
| $\bar{p}$ | | 0.371 | | | 0.413 | |

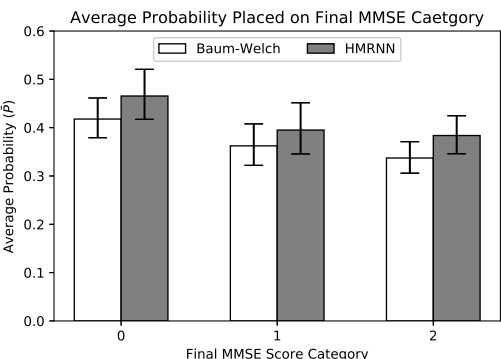

Figure 3: Average probability placed on final MMSE scores, by score category. Recall that the HMRNN's average performance significantly outperforms Baum-Welch (paired $t$-test $p$-value$= 2.396 \times 10^{-6}$). As see in the Figure, this effect is consistent across score categories. Error bars indicate 95% confidence intervals, and do not represent tests for significant differences.

with other neural networks to improve predictive accuracy when additional data is available. In our Alzheimer's case study, augmenting an HMRNN with two predictive networks improves the predictive performance of its parameter solution compared with Baum-Welch. Interestingly, the HMRNN yields a clinically distinct parameter interpretation compared with Baum-Welch, predicting relatively poor diagnostic accuracy for the 'borderline' and 'mild' cognitive impairment states of the MMSE. This suggests that fewer diagnostic categories might improve the MMSE's utility, which is supported by existing MMSE research (e.g., Monroe & Carter, 2012), and suggests the HMRNN might also be used to improve the interpretability of HMM parameter solutions.

In addition to demonstrating the HMRNN's utility in a practical setting, we also make a theoretical contribution by formulating discrete-observation HMMs as a special case of RNNs and by proving coincidence of their likelihood functions. Unlike past approaches, our formulation relies only on matrix multiplication and nonlinear activations, and is designed for generalized use by optimizing for maximum likelihood, which is widely used for HMM training.

Future work may formally assess the time complexity of the HMRNN formulation. The HMRNN converges slower than Baum-Welch for the Alzheimer's case study. Yet packages for training neural networks are designed to efficiently handle large datasets, and therefore the HMRNN may converge faster than Baum-Welch for problems with larger sample sizes. Yet since sequence lengths in healthcare are often considerably shorter than in other domains that employ HMMs (e.g., speech analysis), runtimes will likely be reasonable for many healthcare datasets. We also limited our case study to disease progression; future work might explore the HMRNN in other healthcare domains.

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
