# OpenReview forum: "Hidden Markov models are recurrent neural networks: A disease progression modeling application"
_ICLR.cc/2021/Conference — Reject_

### Official Review · AnonReviewer3 · 2020-10-25
**An interesting connection between RNNs and HMM, but the significance of the connection is not clear enough.**

**Rating:** 5
**Confidence:** 4

**Review:**

### Summary
Authors demonstrated that one can encode the data likelihood function of an HMM using a specialized RNN architecture. Unlike previous work where neurons from different layers were multiplied together, the new encoding strictly followed the classical architecture restrictions of a neural network , i.e. each layer was a weighted sum of the previous layer. Empirically, author showed that the parameter learned by applying gradient descent on the likelihood function is similar to the one that is obtained using the EM algorithm. In addition, authors demonstrated that such formulation enables an application in studying Alzheimer's disease Symptom Progression.

### Strength
* It is an interesting new connection between RNN and HMM that was found by authors.
* I found the Alzheimer's application is very inspiring. Compared to the baseline method that uses a single HMM to model the progression of the Alzheimer for every patient, author proposed to accommodate each individual patient's attributes when predicting the progression. This leads to a more accurate prediction of the final Alzheimer's prediction for each patient.

### Weakness
* The motivation for the connection between RNN and HMM is not very clear, and it hinders the significance of the work. As author has pointed out, there are multiple work that tries to formulate the likelihood function of a graphical model, i.e. a HMM, as a computation graph. Besides the ones that are cited by the author, Chapter 12 of [1] also shows general method to generate a computation graph from a graphical model, and the computation graph evaluates the probability query on the graphical model. These different representations differ only in the syntactical representation of the likelihood function. Although authors managed to formulate the function using the structure of a classical RNN, the underlying function is semantically equivalent to the previous methods. Hence, the gradients w.r.t. the HMM parameters are going to the same in all different representations. Hence, performing gradient descent on a HMRNN does not seem to be different from performing gradient descent on other syntactical representations of the likelihood function.

[1] Modeling and Reasoning with Bayesian Networks. Adnan Darwiche.

### Questions
* It was not clear to me how the supervisions from each patient's attributes were incorporated to model the Alzheimer's progression. As there was no ground truth on the hidden states, I was not sure how the initial probabilities were trained, and how the CDR helped to supervise the training of the hidden states. Are these supervisions incorporated through the loss function?

---

### Official Review · AnonReviewer1 · 2020-10-27
**Lack of experimental supports**

**Rating:** 5
**Confidence:** 3

**Review:**

This paper introduces a novel architecture of recurrent neural network that mimics the working of a standard HMM. In particular, the proposed HMRNN learns model-parameters, which are statistically similar solutions to those of a standard HMM, within a general neural network learning framework. While it is shown the proposed network similarly to the HMM, there are many issues that should be considered critically.

- In Eq. (5), $Y_{t}$ should be $Y_{t}^{n}$ to indicate n-th column/sample and $1_{nx(kc)}$ be $1_{1x(kc)}$.
- The notation of $Psi_{i,j}$ is undefined.
- In the simulation study, it is recommended to initialize parameters with random values, instead of exploiting the background knowledge for the simulation environment.
- It is unclear for the reason of post-processing to compare with the standard HMM.
- The network learning hyper-parameter values need describing for reproducibility.
- Page 7, in the middle: What is the meaning of $h_{t}^{(1)}(2)$?  Due to inconsistency in notations, it is hard to read the manuscript.
- The experimental settings should be denoted in detail.
- Note that the authors argued and stated in Section 3 that HMRNN mimics the operations of the standard HMM and produces statistically similar solutions to the B-W algorithm. If so, it does not make sense to this reviewer why the HMRNN’s performance was superior to the HMM?
- The problem setting over Alzheimer’s disease case study is not clear. To this reviewer’s understanding the input to the models was a sequence of quantized MMSE values and the final output is the category of either CDR>0.5 or CDR<0.5. In Fig. 3, why it is about quantization categories of MMSE, i.e., 0, 1, and 2?
- There should be more rigorous experiments and their results to better support the validity of the proposed method.

---

### Official Review · AnonReviewer4 · 2020-10-27
**A working disease progression model without necessary experimental justifications and baseline comparisons**

**Rating:** 3
**Confidence:** 5

**Review:**

The paper proposes a Hidden Markov Recurrent Neural Network (HMRNN) that mimics the behavior of traditional hidden Markov models (HMM) and shows that the proposed network can be trained to obtain a similar solution as HMM.

Some questions:
1. The topic is not significant. The paper proposes an RNN to mimic the behavior of traditional HMM. Although the paper proves that the model can be optimized to obtain similar parameters as HMM, it doesn't include any comparison or experiment for the usefulness of the model.

2. The paper lacks the necessary discussions. In particular, the paper claims that the proposed model allows “for substantial modularity with other predictive networks”. Even though the paper has added an additional network on the Alzheimer dataset, the paper does not show any ablation experiment on the contribution of each component and eventually the effectiveness of the proposed work.

3. The paper lacks baseline models for the disease progression modeling. There are several disease progression models based on HMM (e.g., [1] and [2]). The paper doesn't compare to such baseline models to demonstrate the effectiveness of the proposed work.

[1] Yu-Ying Liu, Shuang Li, Fuxin Li, Le Song, James M Rehg. Efficientlearningofcontinuous- time hidden markov models for disease progression. In Advances in neural information processing systems, pages 3600–3608, 2015.
[2] Ahmed M Alaa, Scott Hu, and Mihaela van der Schaar. Learning from clinical judgments: Semi-markov-modulated marked hawkes processes for risk prognosis. International Conference on Machine Learning, 2017.

4. The experiment on Alzheimer's disease dataset is not well described (feature dimension, covariances, etc) and no code or pseudo algorithm is provided, making the experimental results and findings hard to be reproduced.

5. The methodology is not well explained. In the traditional HMM, the states in the next timestamp are computed based solely on the state on the current timestamp and are independent of the observation. However, based on equation(4) – (6), the proposed RNN is computing the states based on the current observation.

---

### Official Review · AnonReviewer2 · 2020-10-28
**The paper's claims are much broader than the evidence presented**

**Rating:** 4
**Confidence:** 4

**Review:**

The authors describe an RNN architecture, the HMRNN, which models the log-likelihood in an HMM.  The authors provide theoretical results showing that the HMRNN objective function does, indeed, correspond to the log-likelihood of an HMM.  Synthetic results are presented which compare the learned parameters between the HMRNN and an HMM trained using the Baum-Welch algorithm.  Finally, HMRNN training is augmented and compared to an HMM trained using the Baum-Welch algorithm for the task of Alzheimer's progression prediction.

Firstly, the idea of providing an RNN architecture which models the log-likelihood of an HMM is very interesting.  However, the authors make several important mistakes in the broadness of their claims.  For instance:
"we also make a theoretical
contribution by formulating discrete-observation HMMs as a special case of RNNs and by proving
coincidence of their likelihood functions" <-This does not make the
two models equivalent, which is a major misconception stated
throughout the paper.  For instance, while the authors proved that the
HMM log-likelihood is modeled by the HMRNN, this does not mean HMRNNs
provide equivalent quantities for the beta-recursion, posterior
inference, filtering/smoothing/prediction probabilities, or the
Viterbi path in an HMM.  This is important to distinguish, and I also think the title is incorrect by stating: "HMMs are RNNs."  For the evidence provided in the paper, the following claim would be correct: "RNNs can exactly model the log-likelihood of HMMs."

The empirical evidence, which is meant to support the presented theoretical results showing the equivalence of HMRNNs objective to HMMs log-likelihood, also seem dubious.  The following is claimed in the paper and meant to be supported in Section 4.1: "We demonstrate that an HMRNN trained via gradient descent yields statistically similar solutions to the Baum-Welch algorithm."  However, the "HMRNN trained via gradient descent" is:
"In practice, gradient descent on the HMRNN rarely
yields parameter values that are exactly zero. To facilitate comparability between the HMRNN and
Baum-Welch, we post-process all HMRNN results with one iteration of the Baum-Welch algorithm,
which forces low-probability entries to zero."
Even if the resulting learned parameters between the two were exact, the extra iteration of Baum-Welch added to HMRNN results invalidates the statement that HMRNNs trained via gradient descent produce similar parameters to the Baum-Welch algorithm.  Furthermore, Wasserstein distance is used to compare the two sets of learned parameters, but mean-squared error is much easier to interpret and also the standard measure of similarity when comparing
learned graphical model parameters, e.g., see:
-Noorshams, Nima, and Martin J. Wainwright. "Stochastic belief propagation: A low-complexity alternative to the sum-product algorithm." IEEE Transactions on Information Theory 59.4 (2012): 1981-2000.

In Section 4.1, why is just a single-layer neural network used to
predict the initial state distribution?  Does the Baum-Welch-trained
parameters similarly use this initial state distribution?  If not,
this does not actually fairly compare to the Baum-Welch learned
parameters, since this is something easily included when training an HMM using standard Baum-Welch.  Similarly, the second extension to HMRNN training in section 4.2 does not seem entirely exclusive to RNNs: "Second, at each time point, the probability of being in the most impaired
state... is used to predict concurrent scores on the Clinical Dementia Rating
a global assessment of dementia severity, allowing another clinical
metric to inform estimation." <- This also seems like something that
possible using a very simple, augmented Baum-Welch algorithm for
HMMs.  All the overhead and complexity of HMRNN do not seem warranted.

Also, while the theoretical results do seem to be correct for discrete emission densities, the notation used in the paper is really difficult to understand and the authors are inconsistent in defining variables/notation, making the description of the model difficult to follow along with.  For instance, in definition 1, what is N?  The definition is very confusing without this
definition.  Is N the number of training sequences?  This setup is
also weird, because different training sequences in an HMM may have
different lengths, whereas T is fixed in the paper.   Please state the notation prior to use, i.e., that h_1^{(t)}(j) just a column vector.  Statements involving \mathbf{Y_t} and h_y^{(t)} are also very hard to understand, please consider some other way of describing how the observations are loaded into the model.

Finally, the description of previous work is very light; only one mention of combining ANNs and HMMs is mentioned, and it is largely dismissed that this work used the maximum mutual information training criterion as opposed to maximum likelihood.  However, there is a long history of HMMs
combined with DNNs, which are called hybrid HMMs (there are also
hybrid dynamic Bayesian networks).  These should be discussed and the novelty of the
presented work should be addressed in light of this previous work. Some examples:
-Bourlard, Herv¨¦, and Christian J. Wellekens. "Links between Markov models and multilayer perceptrons." Advances in neural information processing systems. 1989.
-Dahl, George E., et al. "Context-dependent pre-trained deep neural
networks for large-vocabulary speech recognition." IEEE Transactions
on audio, speech, and language processing 20.1 (2011): 30-42.
-Graves, Alex, Navdeep Jaitly, and Abdel-rahman Mohamed. "Hybrid speech recognition with deep bidirectional LSTM." 2013 IEEE workshop on automatic speech recognition and understanding. IEEE, 2013.

Other comments:

It seems warranted that some kind of complexity be given for the
HMRNN, since one of the major selling points of HMMs is that the
forward-recursion can be done in O(T k^2) time and O(T k) memory.

In Section 4.1, please list implementation details regarding how HMRNN
and the Baum-Welch algorithms were run.  This is generally important,
but is even more so since timing numbers are reported in Section 4.2.
Also, since it not discussed
at any length, it is unclear whether the authors ran the
beta-recursion along with the alpha-recursion (the only of the two
discussed), which is necessary to perform the Baum-Welch algorithm.

"Since HMMs may have multiple local optima" <- this statement is
improper.  Training HMMs using EM/Baum-Welch may have multiple local
optima.  Note that other HMM parameter estimation algorithm exist, such as
spectral learning algorithms, which are not subject to multiple local
optima.

It looks like the range of T values in the Alzheimer's application is
3, 4, and 5.  Is that right?  If so, that is incredibly small to be
testing an HMM in general.  While this application is undoubtedly
important, given that the premise is to show that HMRNNs can do things
better than HMMs, a much better benchmark would be for a sufficiently
long time-series, such as speech data.  The authors would probably
gain more traction considering speech data, as:
1.) HMMs have been most leveraged for this application domain
2.) Kaldi exists, and you can easily modify its source
3.) HMMs still play an important part in the training of DNN-speech
architectures, e..g., serving as one of the building blocks in
training a CD-DNN-HMM.
There are many, many other applications of HMMs to much longer time
sequences (with easily modifiable open-source implementations, or the
pomegranate graphical models package is also available) that
the authors can turn to to more convincingly demonstrate the
superiority of their architecture to standard HMMs.

The parantheses around superscripts seems completely unnecessary.

In section 3.3., N is not still not defined, which is confusing.

---

### Decision · Program_Chairs · 2021-01-07
**Final Decision**

**Decision:**

Reject

**Comment:**

There is consensus that the submission is not yet ready for publication. The reviews contain multiple comments and suggestions and I hope they can be useful for the authors.